# Functional Polymorphisms in the Neuropeptide Y (NPY) Gene Associated with Egg Production in Thai Native, Black-Bone, and Commercial Laying Hens Using SNP Markers

**DOI:** 10.3390/ani15050744

**Published:** 2025-03-05

**Authors:** Doungnapa Promket, Jennarong Kammongkun, Jiranan Insee, Wootichai Kenchaiwong, Khanitta Pengmeesri, Thassawan Somchan, Wuttigrai Boonkum

**Affiliations:** 1Branch of Animal Science, Department of Agricultural Technology, Faculty of Technology, Mahasarakham University, Mahasarakham 44150, Thailand; napakran@hotmail.com (D.P.); khanitta.c@msu.ac.th (K.P.); thassawan.s@msu.ac.th (T.S.); 2Applied Animal and Aquatic Sciences Research Unit, Faculty of Technology, Mahasarakham University, Kantarawichai, Mahasarakham 44150, Thailand; 3Bureau of Animal Husbandry and Genetic Improvement, Department of Livestock Development, Pathum Thani 12000, Thailand; jennarongk@yahoo.com; 4Department of Animal Production Technology, Faculty of Agricultural Technology, Kalasin University, Kalasin 46000, Thailand; jiranan.in@ksu.ac.th; 5Small Ruminant Research Unit, Faculty of Veterinary Science, Mahasarakham University, Mahasarakham 44000, Thailand; wootichai.k@msu.ac.th; 6Network Center for Animal Breeding and Omics Research, Khon Kaen University, Khon Kaen 40002, Thailand; 7Department of Animal Science, Faculty of Agriculture, Khon Kean University, Khon Kean 40002, Thailand

**Keywords:** Thai native chickens, egg production, genetics, breeding, heritability

## Abstract

Egg production is a crucial trait in poultry, including native breeds, as it plays a significant role in the global economy. Eggs are a primary source of high-quality protein for human nutrition and remain a central focus of scientific research aimed at improving productivity, maintaining genetic diversity, and ensuring sustainability in poultry farming systems. The integration of molecular technologies, such as single nucleotide polymorphism (SNP) analysis, into native chicken breeding programs has revolutionized genetic improvement by enabling marker-assisted selection (MAS) with high precision. These advancements not only accelerate the improvement of egg production in Thai native breeds but can also be applied to enhance productivity in native breeds from other regions, supporting global efforts in sustainable poultry farming.

## 1. Introduction

Native chickens hold significant cultural, economic, and nutritional value, making them an essential part of local poultry production systems [1]. Native chickens contribute to sustainable farming by utilizing agricultural by-products, reducing food waste, and supporting smallholder livelihoods. Several recent studies highlighted their role in sustainable food systems and their contribution to circular economies within communities worldwide [1,2,3]. Renowned for their unique traits—such as tender meat, high protein content, and exceptional flavor—native chickens are highly valued by consumers [4]. In Thailand, native chickens are the second most popular poultry type after broilers, accounting for 18% of the country’s poultry consumption [5]. Among the native breeds, Pradu Hang Dum, Chee, and black-bone chickens are particularly favored [6,7,8]. These breeds exhibit remarkable adaptability to Thailand’s year-round hot and humid climate and are efficient in foraging and utilizing local feed resources [9]. These characteristics underline their potential for enhancing sustainable livestock production systems and driving genetic improvement programs specifically tailored to native chicken breeds. Thai indigenous chickens face significant challenges in meeting the growing consumer demand due to their relatively low egg production and slower growth rates compared to commercial poultry breeds [10,11]. According to recent studies, native chickens typically produce 80–180 eggs per year, whereas commercial layers can produce 250–320 eggs per year under optimal conditions [3,12,13].

Despite their advantages, such as unique meat quality and strong disease resistance, these limitations hinder their ability to support large-scale production in the poultry industry. Insufficient egg production and slow growth lead to supply shortages, making it difficult to meet market requirements. Enhancing the reproductive performance of these native breeds is therefore critical for ensuring a more sustainable and abundant supply. Efforts have been directed toward improving the egg production traits of Thai indigenous chickens, particularly in promising breeds like Pradu Hang Dum, Chee, and black-bone chickens, through targeted genetic improvement and selective breeding programs.

A sustainable approach to improving egg production in Thai indigenous chickens involves the use of genetic improvement strategies, particularly by selecting individuals with genetic traits associated with higher egg yield. Traditional breeding methods, which rely on phenotypic selection, have been widely used but are often slow and less efficient in achieving genetic progress. Although phenotypic selection can enhance productivity over several generations, significant improvements require considerable time. To address these limitations, integrating molecular techniques, such as marker-assisted selection (MAS), provides a more efficient and faster alternative. MAS enables the identification of specific genetic markers associated with egg production traits, allowing for the targeted selection of chickens with superior genetic potential [14,15]. By applying molecular approaches, breeding programs can accelerate genetic improvement, resulting in higher and more sustainable egg production in Thai indigenous chicken populations.

The use of Single Nucleotide Polymorphism (SNP) technology has become a transformative approach for studying genetic diversity and its influence on economically important traits in poultry, particularly egg production [16,17,18,19]. SNPs, the most common type of genetic variation, result from single base substitutions in the DNA sequence. They serve as powerful genetic markers due to their abundance, stability, and ease of detection. These variations can affect gene function and expression, thereby impacting phenotypic traits such as egg production, egg weight, and shell quality. SNPs are invaluable tools for understanding the genetic makeup of chicken populations and unraveling the genetic basis of complex traits due to their widespread distribution across the genome and high specificity [20]. Integrating SNP analysis into poultry breeding programs has revolutionized genetic improvement by enabling marker-assisted selection (MAS) with high precision [21]. Through SNP-based studies, breeders can identify genetic markers linked to desirable traits, such as increased egg yield and improved egg quality. This facilitates the selection of chickens with superior genetic potential, accelerating breeding progress while promoting sustainable production efficiency. Additionally, SNP analysis aids in assessing genetic diversity within populations, ensuring the long-term genetic health of poultry. By harnessing the advantages of SNP technology, the poultry industry can enhance productivity and meet the growing demand for high-quality eggs, contributing to the sustainability and efficiency of poultry breeding programs.

Neuropeptide Y (NPY) is a highly conserved neuropeptide that plays a critical role in regulating essential physiological processes such as appetite, energy metabolism, and reproductive functions [22]. In poultry, NPY is particularly important for egg production, a key economic trait in the poultry industry. NPY exerts its effects through specific receptors expressed in both the central nervous system and peripheral tissues. In the central nervous system, NPY stimulates feed intake and promotes energy storage, indirectly affecting reproduction by modulating the secretion of gonadotropins and other hormones essential for ovarian function and follicular development [23]. In peripheral tissues, such as the ovaries, NPY contributes to energy homeostasis, ensuring adequate resource allocation for egg production [15,24]. These dual roles highlight the importance of NPY in maintaining metabolic balance and reproductive efficiency for optimal egg yield.

Recent advancements in molecular genetics have demonstrated the potential of the NPY gene as a genetic marker for improving egg production traits in poultry. Single nucleotide polymorphism (SNP) markers, which represent variations in individual nucleotides within the genome, are valuable tools for identifying associations between specific genetic variants and traits of interest. Although the mechanisms through which NPY regulates egg production are not yet fully understood, few studies have explored SNPs within the NPY gene in native chickens to link them with egg production [24,25,26]. The coding region of the NPY gene contains SNPs that could significantly impact egg production traits. Breeding programs can leverage SNP markers within the NPY gene for marker-assisted selection, enabling the identification and breeding of hens with superior genetic potential for egg production. Therefore, this study aimed to address the gap between molecular research and practical genetic improvement by identifying SNP loci within the NPY gene coding sequence and evaluating their association with egg production traits in Thai native chickens. The findings of this study offer valuable genetic tools to enhance egg production efficiency and support the genetic improvement of Thai native chickens. These results also provide insights that can aid in the development of native chicken breeds in other regions worldwide.

## 2. Materials and Methods

All experimental procedures in this study were approved by the Institutional Animal Care and Use Committee (IACUC) of Mahasarakham University, Thailand (Approval No. IACUC-MSU-29/2024).

### 2.1. Experimental Animals and Phenotypic Data Collection

This study included 117 hens from four different breeds: two Thai native breeds (Pradu Hang Dum and Chee), one black-bone breed, and one commercial laying hen breed. There were 59 Pradu Hang Dum (PH), 20 Chee (C), 20 black-bone (BB), and 18 commercial laying hens of the Hy-Line Brown breed (LC) for study (Figure 1). All hens were reared in an open-housing system with a window design, receiving an average of 12 h of natural light per day. The daily air temperature was maintained between 22 and 36 °C, and relative humidity ranged from 50 to 80% throughout the study period. At 16 weeks of age, the hens were transferred to single battery cages (20 × 45 × 40 cm), and egg production was recorded from the first egg laid until 365 days of laying. During the laying period, the hens were provided with 100 g of feed per day (containing 19% crude protein and 2900 kcal ME/kg) and had continuous access to fresh water.

Blood samples were randomly collected from all hens to extract genomic DNA for genetic analysis. Pradu Hang Dum (PH) samples were obtained from the Chiangmai Livestock Research and Breeding Center. Chee (C) and black-bone (BB) samples were collected from the Network Center for Animal Breeding and Omics Research, Faculty of Agriculture, Khon Kaen University. Commercial layer chicken (LC) samples were collected from the Faculty of Agricultural Technology, Kalasin University.

The eight phenotype recorded traits included the age at first egg production (AFEP), the first egg weight (FEW), egg weight at 9 months (EW_9M) and egg weight at 12 months (EW_12M) of egg production period, number of eggs at 9 months (NE_9M) and number of eggs at 12 months (NE_12M) of egg production period, number of eggs per month (EperM), and egg mass (EMs) were analyzed. The total number of eggs was recorded over a 365-day production period. Egg weight was measured using a digital electronic scale with a precision of 0.01 g. These data provided a comprehensive foundation for evaluating the performance of Thai native chickens in this study.

### 2.2. Single Nucleotide Polymorphisms (SNPs) Genotype

#### 2.2.1. DNA Samples, Primer Designing, PCR Amplification

According to Goodwin [27], the guanidine hydrochloride approach was used to separate genomic DNA from whole blood. In this study, guanidine hydrochloride was chosen for DNA extraction because it is highly effective in isolating high-quality DNA, particularly from avian blood samples, which often contain inhibitors. This method ensures efficient cell lysis, protein denaturation, and nucleic acid stabilization, resulting in high DNA yield and purity. Compared to other commonly used methods, guanidine hydrochloride extraction is cost-effective, time-efficient, and well-suited for large-scale sample processing. The blood was briefly mixed with protein precipitation buffer and cell lysis buffer. At 4 °C, the cell lysate was centrifuged for 5 min at 10,000 rpm. Following its transfer to a tube, the supernatant was mixed with 100% isopropanol. The DNA was precipitated for five minutes at 4 °C and 10,000 rpm. Following the disposal of the supernatant, the DNA pellet underwent two rounds of washing in 75% ethanol and dissolved in DNA hydration buffer. Using a Nanodrop 2000c spectrophotometer (Thermo Scientific, Waltham, MA, USA), the DNA quality and concentration were ascertained.

Using online software (https://crm.vazyme.com/cetool/simple.html, accessed on 15 September 2024) and Primer3 BLAST on the NCBI platform, the homology sample was prepared in accordance with the manufacturer’s instructions (TIANGEN, DP304, Beijing, China). Each genotyping experiment was conducted with a total reaction volume of 40 μL for the polymerase chain reaction (PCR). One reaction mixture consisted of the following components: 16.4 μL of nuclease-free water, 0.4 μL of Taq DNA polymerase (Promega, San Diego, CA, USA), 3.2 μL of 50 mM MgCl2, 4 μL of 10× PCR buffer, 4 μL of 1 mM dNTPs, 4 μL of each 5 mM primer (F: CAGAAGGACACGTTGCTGAG and R: GCTTGGATCTGTTTAGCGGG), and 4 μL of genomic DNA at a concentration of 50 ng/μL.

PCR amplification was performed using a thermal cycler (iCycler, BioLab, Reedley, CA, USA; Corbett Research, Mortlake, NSW, Australia, 2003). The process began with an initial denaturation step at 94 °C for 5 min. This was followed by 35 cycles, each including denaturation at 94 °C for 30 s, annealing at the primer-specific temperature 62 °C for 40 s, and extension at 72 °C for 30 s. A final extension step was performed at 72 °C for 5 min. The PCR products were then stored at 4 °C for further analysis.

The amplified products of the NPY genes were analyzed by electrophoresis on a 2% agarose gel. Electrophoresis was conducted at 120 V for 35 min, and the gel was subsequently stained with GELSTAR™ (Gelstar Inc., Patchogue, NY, USA) for 10 min and documented using a Gel Documentation System (Lab Focus Inc., Sterling, VA, USA). This method ensured precise genotype identification.

#### 2.2.2. DNA Sequencing and Bioinformatics Analysis

Sanger and nucleotide sequencing were performed using 40 microliters of the purified PCR product and sequencing analysis software version 7.0 (SqeqA7). ATGC Co. Ltd., Thailand Science Park (TSP) (Pathum Thani, Thailand), carried out the procedure. BLAST tools were used to align the projected gene sequence with the NCBI reference nucleotide sequences of Gallus gallus (http://blast.ncbi.nlm.nih.gov/Blast.cgi; accessed on 15 December 2024). In order to confirm similarities with other species and identify the predicted gene, the predicted nucleotide sequence was also matched to nonredundant nucleotide sequences using BLASTx (nucleotide–nucleotide BLAST). ClustalW was used with BioEdit 7.2 software to carry out multiple alignments based on nucleotide sequences [28].

### 2.3. Statistical Analysis

Before conducting the genetic analysis, the data of eight egg traits from each hen in each chicken breed were verified for normality using the Shapiro–Wilk test and for homogeneity of variance using Levene’s test with the Proc UNIVARIATE procedure in SAS software version 9.4 [29]. Outliers were removed before the analysis. The egg production traits were analyzed using the Proc GLM procedure. The model included fixed effects such as breed and age of hens, along with covariates for environmental factors and management. Differences between chicken breeds were considered significant at a *p*-value of 0.05. The mean ± standard deviation of the data was presented. The methods described by Falconer and Mackay [30] were employed to calculate genotype and allele frequencies, polymorphism information content (PIC), expected heterozygosity (H_E_), and Chi-square test (χ^2^) for examining the Hardy–Weinberg equilibrium (HWE). The least-squares method was used to investigate the relationships between genotype and traits related to egg production (GLM procedure, SAS software version 9.4) [29]. The link function correctly ties the response mean to the linear predictor, according to the GLM assumptions. The following presumptions were made about the model used for data analysis:Y_ij_ = µ + G_i_ + e_ij_
where Y_ij_ is the egg production characteristics, µ is the population mean values for the characteristics, G_i_ is the fixed effects (breed and age of hens) related to the genotype of NPY, e_ij_ is residual error.

## 3. Results

### 3.1. Egg Production Traits in Thai Native Chicken, Black-Bone, and Commercial Laying Hens 

The comparison of egg production traits among four chicken breeds is shown in Table 1. This study highlights significant variations in egg production traits among Thai native chicken breeds and commercial laying hens. The LC breed exhibited superior performance across all traits, with the earliest age at first egg (140.44 ± 5.18 days) and the highest weight of the first egg (43.77 ± 7.05 g). Among native breeds, PH hens had the earliest onset of laying (151.89 ± 9.59 days) and the highest cumulative egg production at NE_9M (159.61 ± 35.49 eggs) and NE_12M (196.90 ± 43.32 eggs), EperM (16.40 ± 3.61 eggs/month) and EMs (42.63 ± 9.82 g/hen/day), significantly higher than other native breeds (*p* < 0.001), though still lower than commercial laying hens. For the C chickens, the cumulative egg production at NE_9M (53.30 ± 23.01 eggs) and NE_12M (73.70 ± 30.70 eggs) was nonsignificant with the BB hens, which recorded 52.85 ± 19.59 eggs (NE_9M) and 87.55 ± 28.78 eggs (NE_12M).

In comparison to native chicken breeds, LC also outperformed them in terms of EperM (21.20 ± 1.17 eggs/month) and EMs (60.67 ± 8.76 g/hen/day) (*p* < 0.001), demonstrating its effectiveness in balancing egg weight and number. PH hens followed with moderate EperM (16.40 ± 3.61 eggs/month) and EM production (42.63 ± 9.82 g/hen/day). Moreover, C and BB hens were less productive, as evidenced by C breed showed EperM and EM production (6.14 ± 2.55 eggs/month; 17.95 ± 5.76 g/hen/day). EperM (7.29 ± 2.39 eggs/month) and EM production (22.06 ± 6.03 g/hen/day) were followed by BB hens (Table 1). LC chickens demonstrated superior performance in all measured traits, particularly in egg production and egg weight, compared to Thai native breeds. Among the native breeds, PH performed best, indicating its potential for further improvement. These results emphasize the importance of selective breeding programs to enhance the productivity of native chicken breeds.

### 3.2. SNP Identification in the Coding Region of the NPY Gene

The analysis of the seven identified SNPs in the coding region of the NPY gene in Thai native chickens (PH and C), BB, and LC breeds is presented in Figure 2. The identified SNPs were as follows: SNP1: g.130 A>G, SNP2: g.301 C>T, SNP3: g.389 G>A, SNP4: g.404 G>T, SNP5: g.427 G>A, SNP6: g.596 G>A, and SNP7: g.647 G>T. These SNP loci were found across all studied breeds, highlighting their potential as genetic markers for further investigation of egg production traits. The identification of SNPs within the coding region of the NPY gene is particularly significant, as NPY plays a crucial role in regulating appetite, energy balance, and reproductive functions in poultry. Variations in these loci may influence egg production traits by modulating hormonal pathways associated with ovarian activity and metabolism. Therefore, these SNPs serve as valuable genetic markers for future research on selective breeding strategies aimed at enhancing the productivity of Thai native chickens and related breeds.

### 3.3. Genotype and Alleles Frequency

Genotype and allele frequency of the NPY gene in four chicken breeds is presented in Table 2. The analysis of genetic variation in chicken breeds was performed by examining the allele and genotype frequencies at 7 SNP loci. This study provides insights into the genetic diversity among different chicken breeds, with a particular focus on the variation in genotype distributions. The genotype and allele frequencies at the SNP1: g.130 A>G reveal significant patterns of genetic variability among the chicken breeds studied. The AA genotype is predominant across all breeds, particularly in the BB breeds, where the frequency is 0.75. This suggests that allele A is strongly favored, with allele frequencies ranging from 0.67 in LC to 0.82 in C. The allele G is less frequent overall, with the highest presence in LC (0.33), further highlighting differences in genetic structure among breeds. These variations in genotype and allele frequencies provide insights into the genetic differentiation and potential selection pressures influencing these populations. The analysis of genotype and allele frequencies for SNP2: g.301 C>T reveals notable genetic variation among chicken breeds. The CC genotype is predominant across most populations, particularly in the LC breed (0.78), suggesting a preference for the C allele (0.68–0.81) in these populations. Conversely, the CT and TT genotypes are less frequent. The allele frequencies indicate that allele C is more common overall, with allele T being less prevalent, particularly in the LC breed (0.19). The genotype and allele frequencies for SNP3: g.389 G>A show in the PH breed; the GG genotype is most frequent (0.46), with a balanced allele frequency of G (0.53) and A (0.47). The C breed chickens have equal genotype frequencies for GG and GA, both at 0.35, while the genotype frequency for AA is 0.30. However, the allele frequencies for G and A are nearly identical, at 0.52 and 0.48, respectively. In contrast, the BB breed shows the opposite trend compared to other breeds, with the genotype frequency for AA being the highest at 0.55, while the GA genotype has the lowest frequency at 0.10. Nevertheless, the allele frequencies for G and A in the BB breed are also close, at 0.40 and 0.60, respectively. The LC breed shows a relatively balanced genotype frequency of GG, GA, and AA at 0.44, 0.22, and 0.33, respectively. The allele frequencies for G and A are also nearly identical, at 0.56 and 0.44, respectively. The allele frequency of G is found to be similar to that of allele A across all chicken breeds, indicating a balanced genetic distribution at the studied loci. Three genotypes (GG, GT, and TT) were found at the SNP4: g.404 G>T locus; the GG genotype was the most prevalent, and the allele frequency of G was higher than T in all chicken breeds. The chicken breeds PH, C, BB, and LC have genotype frequency for GG, with values of 0.53, 0.70, 0.65, and 0.56, respectively. The allele frequency of G in PH, C, BB, and LC being 0.54, 0.78, 0.70, and 0.67, respectively. At SNP5: g.427 G>A, three genotypes were observed: GG, GA, and AA, with the AA genotype having a higher frequency than the GG genotype, except in the BB breed, where the GG genotype (0.45) slightly exceeds the AA genotype (0.40). The allele frequency of A in the PH, C, and LC breeds (0.53, 0.53, and 0.64, respectively) is higher than that of allele G (0.47, 0.47, and 0.36, respectively). In contrast, the allele frequencies of G and A in the BB breed are 0.52 and 0.48, respectively. At SNP6: g.596 G>A, a base substitution from G to A was observed. Allele frequency analysis revealed that allele A was more frequent than allele G across all breeds, except for the LC breed. The AA genotype showed the highest frequency in the native breeds PH, C, and BB, with frequencies of 0.46, 0.50, and 0.75, respectively, the allele frequency of A in the PH, C, and BB breeds was 0.55, 0.70 and 0.85, respectively. LC showing allele G (0.61) had a higher frequency than allele A (0.39). SNP7: g.647 G>T, a base substitution from G to T, was identified. The GG genotype exhibited the highest frequency across all breeds (PH, C, BB, and LC), ranging from 0.56 to 0.70. In contrast, the recessive homozygous genotype (TT) had the lowest frequency, ranging from 0.05 to 0.19. Allele frequency analysis revealed that allele G was more frequent than allele T in all chicken breeds studied (Table 2). The SNP loci in the NPY gene exhibit distinct allele and genotype frequency patterns across the chicken breeds, with some loci showing predominant alleles (allele A for SNP1, allele C for SNP2, allele G for SNP4, and allele G for SNP7) and others showing more balanced allele distributions (SNP3 and SNP5).

### 3.4. Genetic Diversity

Table 2 shows the genetic diversity of chicken breeds, as assessed through SNP genotype and allele frequency data. Key indicators, including allele frequency, genotype frequency, Chi-square (X^2^) values, polymorphism information content (PIC), and expected heterozygosity (H_E_), provide a comprehensive understanding of the genetic structure of these breeds. The distribution of genotype frequencies across SNP loci further underscores differences in genetic diversity. Certain loci, such as SNP1: g.130 A>G, SNP2: g.301 C>T, SNP4: g.404 G>T, and SNP7: g.647 G>T, exhibit a predominance of homozygous genotypes (AA, CC, GG, and GG, respectively), indicating lower heterozygosity and potentially reduced genetic variation in these populations. However, loci like SNP3: g.389 G>A and SNP5: g.427 G>A display more balanced genotype frequencies, indicative of greater genetic diversity

The analysis of SNP loci reveals that C breed is in Hardy–Weinberg equilibrium (HWE) for most loci, except at SNP4: g.404 G>T. Similarly, the LC breed exhibits HWE at SNP position SNP3: g.389 G>A, SNP4: g.404 G>T, SNP5: g.427 G>A, SNP6: g.596 G>A, and SNP7: g.647 G>T, suggesting no significant deviation from expected genotype frequencies at these loci. The BB breed shows genetic equilibrium at loci SNP6: g.596 G>A and SNP7: g.647 G>T, indicating stable allele distributions within the population. In contrast, the PH breed does not conform to HWE. Additionally, PIC values, which measure the informativeness of genetic markers, vary across loci. The PIC values obtained in this study range from 0.22 to 0.50, indicating moderate genetic diversity within the populations. Moderate PIC values like these are beneficial for understanding the genetic makeup of the populations and for utilizing these markers in selection strategies aimed at improving egg productivity. The seven SNP loci in this study had expected heterozygosity (He) values ranging from 0.26 to 0.50, meaning that the populations had a moderate level of genetic diversity.

### 3.5. Associations Between Seven SNP Polymorphisms and Chicken Egg Production

The relationship between seven SNP loci and egg production traits in chickens is summarized in Table 3. SNP1 (g.130 A>G) and SNP2 (g.301 C>T) were associated with age at first egg production (AFEP). Chickens with the GG genotype at SNP1 had an AFEP of 150.21 days, while the AG genotype showed a slightly later AFEP (158.82 days), though the difference was not significant. At SNP2, the CC genotype exhibited the best AFEP (157.10 days), but this was not significantly different from t_he_ TT genotype (159.92 days). These loci may influence the onset of egg production, offering potential for reproductive trait improvement in breeding programs.

At SNP3 (g.389 G>A), the GG genotype showed the highest values for number of eggs at 12 months (NE_12M) and number of eggs per month (EperM), with 183.51 and 15.29 eggs, respectively, though these values were not significantly different from the GA genotype. SNP4 (g.404 G>T) was significantly associated with AFEP and first egg weight (FEW). Hens with the TT genotype had the lowest AFEP (156.23 days), whereas those with the GT genotype had the highest AFEP (168.63 days) (*p* < 0.05). For the FEW trait, hens with the GT genotype had the highest FEW compared to other genotypes (*p* < 0.05). SNP5 (g.427 G>A) and SNP6 (g.596 G>A) were linked to multiple traits, including egg weight at 9 and 12 months (EW_9M and EW_12M), number of eggs (NE_9M and NE_12M), number of eggs per month (EperM), and egg mass (EMs). At SNP5, the GA genotype had the highest EW_9M (46.78 g) and EW_12M (48.76 g), while the AA genotype exhibited superior NE_9M, NE_12M, and EperM (139.57, 180.25, and 15.02 eggs, respectively) and the highest EMs (40.69 g/egg/hen). SNP6 revealed similar trends, with the GG genotype outperforming others in EW_9M, EW_12M, NE_9M, NE_12M, EperM, and EMs. Finally, no significant associations were found between SNP7 (g.647 G>T) and any egg production traits.

## 4. Discussion

The results of this study reveal notable differences in egg production traits between Thai native chicken breeds and commercial laying hens, highlighting significant genetic and phenotypic variations among the studied populations. The superior performance of the commercial laying hens (LC) across all traits—such as earlier age at first egg production, higher cumulative egg production, and more efficient egg mass production—underscores their genetic potential, which is likely a result of extensive selective breeding in commercial laying hens.

Among Thai native breeds, the Pradu Hang Dum (PH) chickens exhibited higher productivity than the Chee (C) chickens, particularly in number of eggs at 9 and 12 months (NE_9M, NE_12M) of the egg production period, number of eggs per month (EperM), and egg mass (EMs) traits. This difference can be attributed to the fact that PH chickens have been genetically selected for egg production traits over a period of more than 15 years, leading to improved genetic potential for these traits. In contrast, C chickens have primarily been raised for genetic conservation, without any explicit selection for egg production traits. Consequently, egg production traits in C chickens have not been as effectively selected or improved. Additionally, differences in environmental conditions and management practices also influenced egg production.

However, while PH chickens outperformed other native breeds, their productivity remained significantly lower than that of LC chickens, reflecting limited genetic progress in native breeds compared to commercial laying hens. Despite their relatively low productivity, the Chee (C) and black-bone (BB) native chicken breeds hold significant economic value and cultural importance, particularly in regions like Thailand, Hong Kong, Southern China, and Japan [7]. Lower productivity in native breeds may stem from genetic limitations [31], but research has shown that crossing exotic breeds with native chickens can significantly improve performance, particularly in growth and egg production traits, while maintaining a desirable meat quality [11,14,32]. Studies also demonstrate that native chickens exhibit strong genetic potential for growth and reproductive improvements, with favorable heritability estimates and genetic correlations when analyzed using appropriate statistical methods [33]. Implementing genetic improvement strategies for native chickens—such as selecting for traits like earlier egg-laying onset, higher cumulative egg production, and greater egg mass efficiency—can enhance the economic viability of these breeds while preserving their unique genetic and cultural heritage [9,34,35]. Furthermore, integrating advanced genetic tools, such as marker-assisted selection and genomic analyses, into breeding programs can accelerate genetic progress in Thai native chickens. This study provides a detailed analysis of genetic variation and diversity among Thai native chicken breeds and commercial laying hens by examining allele and genotype frequencies at seven SNP loci in the neuropeptide Y (NPY) gene. While certain SNP loci (SNP1, SNP2, SNP4, and SNP7) exhibited predominantly homozygous genotypes, indicating lower heterozygosity and potentially reduced genetic diversity, other loci (e.g., SNP3 and SNP5) showed more balanced genotype frequencies, suggesting greater genetic diversity. Monitoring and maintaining heterozygosity levels are critical for ensuring adaptability and resilience in these populations.

The moderate PIC values (0.22–0.50) observed in this study demonstrate the informativeness of the SNP markers in distinguishing genetic variants among chicken populations. These values suggest that the loci analyzed provide valuable genetic information for breeding programs, particularly for enhancing egg productivity. This finding aligns with the study by Promket et al. [36], which reported moderate PIC values in Thai indigenous chickens. According to Ding et al. [37], loci with a PIC < 0.25 are modestly informative, those with a PIC < 0.50 are reasonably informative, and those with a PIC > 0.50 are highly informative. The heterozygosity within populations, estimated using the expected heterozygosity (He), reflects the genetic diversity. The He values across all breeds at seven SNPs ranged from 0.36 to 0.50, indicating intermediate genetic diversity. These values suggest a reduced risk of inbreeding or mutations as the populations maintain a respectable level of genetic variation. Although the moderate heterozygosity levels indicate that the populations are not immediately at risk of genetic erosion, continuous monitoring and management of genetic diversity are essential to preserve the overall genetic health of these populations. These genetic diversity metrics provide critical insights into the variation and genetic structure of chicken populations, with significant implications for breeding programs aimed at improving production traits while maintaining genetic diversity. High genetic diversity (He > 0.5) is associated with greater adaptability, whereas lower diversity (He < 0.5) may indicate genetic vulnerabilities [17].

The Hardy–Weinberg equilibrium (HWE) analysis further revealed differences in genetic structure among breeds. While some populations, such as the LC breed, were in genetic equilibrium at most loci, others, such as the PH breed, deviated from HWE. These deviations may be attributed to selection pressures, genetic drift, or historical management practices influencing allele frequencies. Factors such as recombination rates, selection, mating systems, genetic linkage, and population structure contribute to HWE deviations [38]. The results underscore the importance of maintaining genetic diversity to ensure the sustainability of chicken populations. The genetic variability observed in native chicken breeds offers opportunities for targeted breeding strategies to improve economically important traits, such as egg productivity and disease resistance, while preserving the unique genetic characteristics of these populations. However, continuous genetic monitoring and management are critical to ensuring long-term adaptability and productivity. This study also explored the genetic associations between specific SNP polymorphisms in the NPY gene and various egg production traits in chickens. The NPY gene, which regulates energy homeostasis, provides insights into feed conversion efficiency and reproductive performance [39]. Neuropeptide Y (NPY) modulates key neuroendocrine pathways in the brain, particularly in the paraventricular nucleus (PVN), influencing feeding behaviors, body weight, and ovarian activity during different reproductive stages [22].

Notable findings include the associations of SNP1: g.130 A>G and SNP2: g.301 C>T with Age at First Egg Production (AFEP). Although differences in AFEP among genotypes were not statistically significant, the trends suggest biological relevance, especially when combined with other selection strategies. SNP3: g.389 G>A was associated with the Number of Eggs at 12 Months (NE_12M) and Number of eggs per month (EperM), with the GG genotype exhibiting superior performance in egg production rates. SNP4: g.404 G>T presented interesting associations, where the TT genotype correlated with the best AFEP, while the GT genotype demonstrated the highest First Egg Weight (FEW). However, the lower FEW observed in the TT genotype suggests potential trade-offs when prioritizing traits for selection. SNP5: g.427 G>A and SNP6: g.596 G>A were linked to multiple egg production traits, such as egg weight, number of eggs, and egg mass, making these loci particularly promising for selective breeding programs. Conversely, SNP7: g.647 G>T showed no significant associations with any traits, underscoring the complexity of genetic contributions to phenotypic variation. This study highlights the utility of SNP1, SNP2, SNP3, SNP4, SNP5, and SNP6 as potential markers for selective breeding to enhance reproductive performance and productivity in chickens. According to previous studies, Chen et al. [40] identified 12 significant SNPs associated with Age at First Egg (AFE) in layer breeds, which showed strong correlations with body weight and first egg weight. However, there were no overlapping genes between breeds, indicating distinct genetic foundations for AFE. In addition, Fu et al. [41] reported 11 genome-wide significant SNPs related to egg production traits, including age at first egg and egg number. Notably, SNPs in the NEO1, ADPGK, and CYP11A1 genes were linked to egg number, which could aid in selective breeding strategies for chickens. Zhou et al. [42] identified nine significant SNP loci associated with egg production traits in Peking ducks, with SNPs linked to age at first egg and egg number. These SNPs showed varying correlations with traits, suggesting their potential for use in selective breeding strategies. Furthermore, other studies have highlighted the influence of the NPY gene on egg production in Indigenous Noi chickens [26].

Moreover, the NPY gene, located on chromosome 2 in chickens and composed of 36 amino acids, has been reported to influence productive traits in indigenous chickens of Iraq [20] and egg production traits in native chickens of India [25]. Differences in the distribution of NPY neurons in the PVN across reproductive stages further emphasize the gene’s role in regulating egg production [22]. This study provides valuable insights that contribute to the improvement of poultry breeding programs by incorporating genetic markers to enhance productivity while preserving the unique traits of native chicken breeds. The genetic markers identified in this study could be integrated into Marker-Assisted Selection (MAS) programs, allowing for the early selection of hens with superior egg production traits. This approach would improve breeding efficiency, shorten generation intervals, and enhance the productivity of Thai native chickens while maintaining their genetic diversity.

In addition, by utilizing the genetic diversity within native populations, breeding programs can select traits that improve egg production, disease resistance, and adaptability to local conditions, leading to more efficient and sustainable poultry farming. Furthermore, such programs help conserve native breeds by ensuring their continued relevance in agricultural systems, providing both economic benefits and the protection of valuable genetic resources. This approach aligns with the dual goal of improving productivity while safeguarding the genetic heritage of native poultry breeds. Future research should validate these findings in larger, genetically diverse populations and explore potential interactions between SNPs and environmental factors. Integrating these genetic markers into genomic selection programs can improve the precision and efficiency of breeding efforts, ultimately enhancing productivity and sustainability in poultry production systems.

## 5. Conclusions

Among the native breeds studied, the Pradu Hang Dum breed showed the earliest onset of laying and the highest cumulative egg production at both 9 and 12 months of age. The research also identified significant genetic diversity across the chicken breeds, with moderate heterozygosity observed at several single nucleotide polymorphism (SNP) loci on the neuropeptide Y (NPY) gene. Several SNP loci were associated with important egg production traits, including age at first egg production (AFEP), first egg weight (FEW), egg weight at 9 and 12 months (EW_9M and EW_12M), number of eggs at 9 and 12 months (NE_9M and NE_12M), number of eggs per month (EperM), and egg mass (EMs). These findings highlight the potential to improve egg production in Thai native chickens through selective breeding and genetic marker-assisted selection. This study offers valuable insights for designing breeding programs aimed at increasing productivity while maintaining the unique characteristics of native breeds. It emphasizes the importance of leveraging genetic diversity to develop sustainable poultry production systems.

## Figures and Tables

**Figure 1 animals-15-00744-f001:**
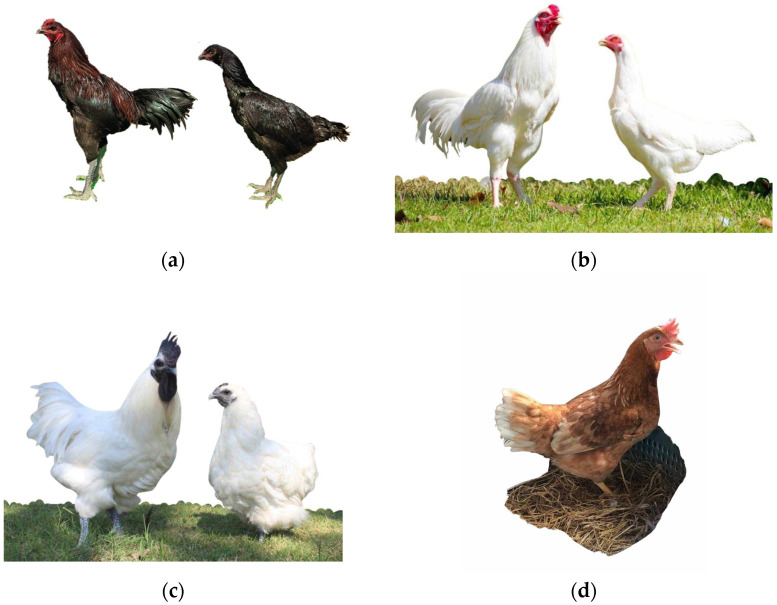
Characteristics of (**a**) Pradu Hang Dum, (**b**) Chee, (**c**) black-bone, and (**d**) commercial laying chickens studied.

**Figure 2 animals-15-00744-f002:**
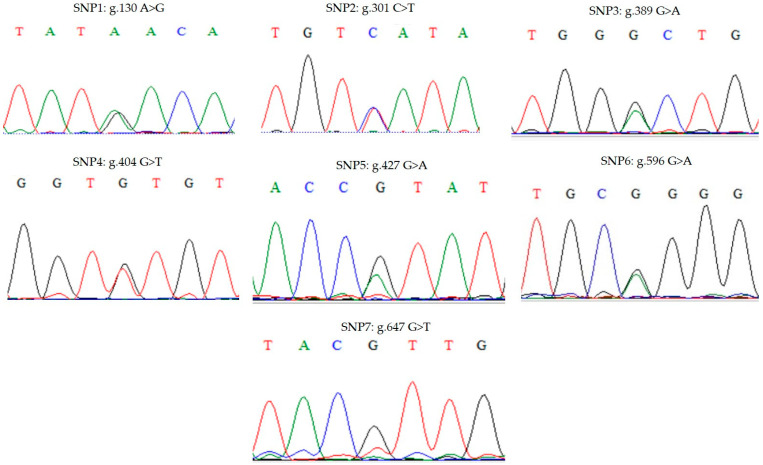
SNPs in the coding region of the NPY gene in Thai native chicken, black-bone, and commercial laying hens.

**Table 1 animals-15-00744-t001:** Average ± standard deviation of egg production traits for Thai native, black-bone, and commercial laying hen breeds.

Traits	Chicken Breeds				*p*-Value
	PH	C	BB	LC	
AFEP, day	151.89 ± 9.59 ^c^	188.05 ± 24.14 ^a^	170.60 ± 12.41 ^b^	140.44 ± 5.18 ^d^	<0.001
FEW, g	33.29 ± 6.07 ^b^	35.08 ± 3.86 ^b^	41.55 ± 1.05 ^a^	43.77 ± 7.05 ^a^	<0.001
EW_9M, g	44.81 ± 3.10 ^b^	41.15 ± 2.94 ^c^	44.45 ± 1.09 ^b^	51.48 ± 6.91 ^a^	<0.001
EW_12M, g	44.95 ± 3.06 ^c^	43.30 ± 3.16 ^c^	48.55 ± 1.05 ^b^	53.50 ± 6.91 ^a^	<0.001
NE_9M, egg	159.61 ± 35.49 ^b^	55.30 ± 23.01 ^c^	52.85 ± 19.59 ^c^	195.61 ± 15.79 ^a^	<0.001
NE_12M, egg	196.90 ± 43.32 ^b^	73.70 ± 30.70 ^c^	87.55 ± 28.78 ^c^	254.44 ± 14.08 ^a^	<0.001
EperM, egg	16.40 ± 3.61 ^b^	6.14 ± 2.55 ^c^	7.29 ± 2.39 ^c^	21.20 ± 1.17 ^a^	<0.001
EMs, g/hen/day	42.63 ± 9.82 ^b^	17.95 ± 5.76 ^c^	22.06 ± 6.03 ^c^	60.67 ± 8.76 ^a^	<0.001

PH: Pradu Hang Dum; C: Chee; BB: black-bone; LC: commercial laying hens; AFEP: age at first egg production; FEW: first egg weight; EW_9M: egg weight at 9 months; EW_12M: egg weight at 12 months; NE_9M: number of eggs at 9 months; NE_12M: number of eggs at 12 months; EperM: number of eggs per month; EMs: egg mass. ^a,b,c,d^ = different superscript letters indicate significant differences within the row for each parameter at *p* < 0.05.

**Table 2 animals-15-00744-t002:** Genotype and allele frequency of NPY gene in Thai native, black-bone, and commercial laying hen breeds.

SNPs	Chicken Breeds	Genotype Frequency (n)			Allele Frequency		PIC	H_E_	Chi-Square
SNP1: g.130 A>G		AA	AG	GG	A	G			
	PH	0.69 (41)	0.14 (8)	0.17 (10)	0.76	0.24	0.30	0.36	23.08
	C	0.70 (14)	0.25 (5)	0.05 (1)	0.82	0.175	0.25	0.29	0.36
	BB	0.75 (15)	0.10 (2)	0.15 (3)	0.80	0.20	0.27	0.32	9.45
	LC	0.61 (11)	0.11 (2)	0.28 (5)	0.67	0.33	0.35	0.44	10.13
	All	0.69 (81)	0.15 (17)	0.16 (19)	0.76	0.24	0.29	0.36	41.55
SNP2: g.301 C>T		CC	CT	TT	C	T			
	PH	0.63 (37)	0.12 (7)	0.25 (15)	0.69	0.31	0.34	0.43	30.96
	C	0.50 (10)	0.35 (7)	0.15 (3)	0.68	0.32	0.34	0.44	0.82
	BB	0.65 (13)	0.05 (1)	0.30 (6)	0.68	0.32	0.34	0.44	15.70
	LC	0.78 (14)	0.06 (1)	0.17 (3)	0.81	0.19	0.26	0.31	12.18
	All	0.63 (74)	0.14 (16)	0.23 (27)	0.70	0.30	0.33	0.42	53.13
SNP3: g.389 G>A		GG	GA	AA	G	A			
	PH	0.46 (27)	0.15 (9)	0.39 (23)	0.53	0.47	0.37	0.50	28.38
	C	0.35 (7)	0.35 (7)	0.30 (6)	0.52	0.48	0.37	0.50	1.78
	BB	0.35 (7)	0.10 (2)	0.55 (11)	0.40	0.60	0.36	0.48	12.53
	LC	0.44 (8)	0.22 (4)	0.33 (6)	0.56	0.44	0.37	0.49	5.45
	All	0.42 (49)	0.19 (22)	0.39 (46)	0.51	0.49	0.37	0.50	45.51
SNP4: g.404 G>T		GG	GT	TT	G	T			
	PH	0.53 (31)	0.03 (2)	0.44 (26)	0.54	0.46	0.37	0.50	30.35
	C	0.70 (14)	0.15 (3)	0.15 (3)	0.78	0.23	0.29	0.35	6.50
	BB	0.65 (13)	0.10 (2)	0.25 (5)	0.70	0.3	0.33	0.42	11.61
	LC	0.56 (10)	0.22 (4)	0.22 (4)	0.67	0.33	0.35	0.44	4.50
	All	0.58 (68)	0.09 (11)	0.32 (38)	0.63	0.37	0.36	0.47	74.64
SNP5: g.427 G>A		GG	GA	AA	G	A			
	PH	0.42 (25)	0.08 (5)	0.49 (29)	0.47	0.53	0.37	0.50	40.62
	C	0.30 (6)	0.35 (7)	0.35 (7)	0.47	0.53	0.37	0.50	1.78
	BB	0.45 (9)	0.15 (3)	0.40 (8)	0.52	0.48	0.37	0.50	9.78
	LC	0.22 (4)	0.28 (5)	0.50 (9)	0.36	0.64	0.35	0.46	2.85
	All	0.38 (44)	0.17 (20)	0.45 (53)	0.46	0.54	0.50	0.50	50.36
SNP6: g.596 G>A		GG	GA	AA	G	A			
	PH	0.36 (21)	0.19 (11)	0.46 (27)	0.45	0.55	0.37	0.49	22.92
	C	0.10 (2)	0.40 (8)	0.50 (10)	0.30	0.70	0.33	0.42	0.05
	BB	0.05 (1)	0.20 (4)	0.75 (15)	0.15	0.85	0.22	0.26	0.93
	LC	0.50 (9)	0.22 (4)	0.28 (5)	0.61	0.39	0.36	0.48	5.10
	All	0.28 (33)	0.23 (27)	0.49 (57)	0.40	0.60	0.36	0.48	31.42
SNP7: g.647 G>T		GG	GT	TT	G	T			
	PH	0.58 (34)	0.24 (14)	0.19 (11)	0.69	0.31	0.33	0.42	11.44
	C	0.65 (13)	0.20 (4)	0.15 (3)	0.75	0.25	0.30	0.38	4.36
	BB	0.70 (14)	0.25 (5)	0.05 (1)	0.82	0.18	0.25	0.29	0.36
	LC	0.56 (10)	0.39 (7)	0.06 (1)	0.75	0.25	0.30	0.38	0.02
	All	0.61 (71)	0.26 (30)	0.14 (16)	0.74	0.26	0.31	0.39	13.66

PH: Pradu Hang Dum; C: Chee; BB: black-bone; LC: commercial laying hens; PIC: polymorphism information content; H_E_: expected heterozygosity.

**Table 3 animals-15-00744-t003:** The association between SNP loci in the NPY gene and egg production traits in Thai native, black-bone and commercial laying hen breeds.

SNPs	Genotype	AFEP	FEW	EW_9M	EW_12M	NE_9M	NE_12M	EperM	EMs
SNP1: g.130 A>G	AA	161.84 ^a^	36.56	45.12	46.60	123.26	160.54	13.38	36.70
	AG	158.82 ^ab^	37.14	45.64	47.05	132.47	164.59	13.71	37.97
	GG	150.21 ^b^	37.17	44.84	46.26	150.79	190.58	15.88	41.52
*p*-value		0.02	0.88	0.64	0.61	0.32	0.57	0.57	0.89
SNP2: g.301 C>T	CC	157.10 ^b^	37.04	45.56	47.06	132.41	168.05	14.00	38.45
	CT	169.93 ^a^	37.13	44.25	45.54	120.38	150.81	12.56	34.75
	TT	159.92 ^ab^	35.69	44.55	46.01	125.07	169.41	14.11	37.27
*p*-value		0.04	0.35	0.17	0.06	0.95	0.96	0.96	0.85
SNP3: g.389 G>A	GG	157.04	36.47	44.53	45.89	140.47	183.51 ^a^	15.29 ^a^	40.40
	GA	163.63	37.12	45.52	46.92	127.23	162.32 ^ab^	13.52 ^ab^	38.19
	AA	160.17	36.85	45.63	47.23	117.80	149.13 ^b^	12.42 ^b^	34.52
*p*-value		0.35	0.40	0.38	0.07	0.06	0.03	0.03	0.21
SNP4: g.404 G>T	GG	159.86 ^ab^	37.37 ^ab^	44.71	46.40	126.99	165.66	13.80	37.31
	GT	168.63 ^b^	40.63 ^a^	46.76	48.39	115.00	152.45	12.70	36.56
	TT	156.23 ^a^	34.48 ^b^	45.47	46.46	136.87	170.55	14.21	38.64
*p*-value		0.02	<0.01	0.18	0.09	0.32	0.32	0.32	0.41
SNP5: g.427 G>A	GG	161.18	35.67	44.16 ^b^	45.47 ^b^	118.93 ^b^	150.39 ^b^	12.53 ^b^	33.34 ^b^
	GA	161.50	38.40	46.78 ^a^	48.76 ^a^	123.55 ^ab^	162.65 ^ab^	13.55 ^ab^	19.20 ^c^
	AA	157.37	37.00	45.35 ^ab^	46.74 ^ab^	139.57 ^a^	180.25 ^a^	15.02 ^a^	40.69 ^a^
*p*-value		0.51	0.17	<0.01	<0.01	0.03	0.01	0.01	<0.01
SNP6: g.596 G>A	GG	151.21 ^b^	36.54	46.44 ^a^	47.34 ^a^	152.33 ^a^	189.42 ^a^	15.78 ^a^	42.86 ^a^
	GA	162.88 ^a^	36.68	45.07 ^ab^	46.59 ^ab^	123.48 ^b^	158.81 ^ab^	13.23 ^ab^	36.51 ^ab^
	AA	162.71 ^a^	36.89	44.44 ^b^	46.19 ^b^	118.25 ^b^	155.86 ^b^	12.98 ^b^	35.21 ^b^
*p*-value		<0.01	0.33	<0.01	0.03	<0.01	<0.01	<0.01	<0.01
SNP7: g.647 G>T	GG	160.59	37.10	45.10	46.63	120.04	153.21	12.76	34.90
	GT	156.93	37.19	46.03	47.42	143.20	187.73	15.64	42.90
	TT	155.56	34.31	43.69	45.00	142.63	182.06	15.17	40.14
*p*-value		0.63	0.28	0.31	0.34	0.36	0.24	0.23	0.14

AFEP: Age at first egg production; FEW: first egg weight; EW_9M: egg weight at 9 months; EW_12M: egg weight at 12 months; NE_9M: number of eggs at 9 months; NE_12M: number of eggs at 12 months; EperM: number of eggs per month; EMs: egg mass. ^a,b,c^ = different superscript letters indicate significant differences within column for each parameter at *p* < 0.05.

## Data Availability

The original contributions presented in this study are included in the article; further inquiries can be directed at the corresponding author.

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
