# Peer review of "Functional Polymorphisms in the Neuropeptide Y (NPY) Gene Associated with Egg Production in Thai Native, Black-Bone, and Commercial Laying Hens Using SNP Markers"

_animals, 2025, doi:10.3390/ani15050744_

Round 1
Reviewer 1 Report
Comments and Suggestions for Authors
I consider the research topic presented in the paper "Functional Polymorphisms in the Neuropeptide Y (NPY) Gene Associated with Egg Production in Thai Native, Black-bone and Commercial Layers Using SNP Markers" (manuscript ID: animals-3474420) to be original and relevant to the field.
The references are appropriate. It can be improved
Line 198 - missing reference [26],
Lines 112 and 115 can be improved – use single citation of publications [21].
The conclusions are consistent with the presented evidence and arguments and address the main question posed. The work brings new elements to the current state of knowledge.
The preservation of native poultry breeds is of great importance. Native poultry breeds are perfectly adapted to harsh environmental conditions. They are morphologically and productionally diverse, provide products, i.e., meat and eggs, with unique dietary and taste value and are invaluable genetic resources that can be used in future genetic improvement programs for commercial chicken flocks. The analysis of genetic and production parameters of hen strains can be a valuable source of information for the scientific community dealing with animal biodiversity. Phenotypic selection can enhance productivity over several generations, significant improvements require considerable time. Integrating molecular techniques provides a more efficient and faster alternative.
Author Response
Dear Reviewer1,
Response to Reviewer 1 Comments
I consider the research topic presented in the paper "Functional Polymorphisms in the Neuropeptide Y (NPY) Gene Associated with Egg Production in Thai Native, Black-bone and Commercial Layers Using SNP Markers" (manuscript ID: animals-3474420) to be original and relevant to the field.
We sincerely appreciate the reviewer’s positive feedback and their recognition of the improvements made in the manuscript. We hope that the revised manuscript will meet your expectations. Our responses to each comment are listed below.
The references are appropriate. It can be improved
Point 1: Line 198 - missing reference [26],
Response 1: We added reference number 29 to this section. Please see line 207.
Point 2: Lines 112 and 115 can be improved – use single citation of publications [21].
Response 2: We revised the manuscript to use a single citation, and the reference order was updated to [23]. Please see line 122.
Point 3: The conclusions are consistent with the presented evidence and arguments and address the main question posed. The work brings new elements to the current state of knowledge.
The preservation of native poultry breeds is of great importance. Native poultry breeds are perfectly adapted to harsh environmental conditions. They are morphologically and productionally diverse, provide products, i.e., meat and eggs, with unique dietary and taste value and are invaluable genetic resources that can be used in future genetic improvement programs for commercial chicken flocks. The analysis of genetic and production parameters of hen strains can be a valuable source of information for the scientific community dealing with animal biodiversity. Phenotypic selection can enhance productivity over several generations, significant improvements require considerable time. Integrating molecular techniques provides a more efficient and faster alternative.
Response 3: We sincerely appreciate the reviewer’s thoughtful comments on the importance of preserving native poultry breeds and their role in genetic improvement programs. We fully agree that native breeds are valuable genetic resources, particularly due to their adaptability to harsh environments and their unique production qualities. Thank you once again for your valuable input.
Best Regards,
Wuttigrai Boonkum

Reviewer 2 Report
Comments and Suggestions for Authors
This study investigates the role of single nucleotide polymorphisms (SNPs) in the neuropeptide Y (NPY) gene and their association with egg production traits in Thai native, black-bone, and commercial layer chickens. The study aims to identify genetic markers that could be applied in marker-assisted selection (MAS) to improve the productivity of native chicken breeds.
The study is well-structured, presenting a clear research question and appropriate methodology. However, some areas require further clarification and revision.
The abstract effectively summarizes the study but could better highlight the practical applications of these findings in real-world poultry breeding programs. For example, the statement "These findings highlight the potential of using genetic markers for improving egg production traits in Thai native chickens" (lines 46–48) should be expanded to explain how such genetic markers could be implemented in MAS programs in the discussion part.
The introduction provides relevant background information on the economic and cultural significance of native chickens.
However, lines 55–56 mention the role of native chickens in circular economies without citing recent studies supporting this claim.
Lines 66–68 should include actual production statistics for native chickens to provide a more quantitative comparison with commercial layers.
The materials and methods section is generally well-detailed but lacks specific information on some key aspects.
Lines 144–145 mention that blood samples were randomly collected, but it is unclear whether they were obtained from different farms or a single research station.
Lines 157–163: the DNA extraction method follows a standard protocol, but the authors should justify why guanidine hydrochloride was used instead of other commonly used DNA extraction methods.
Lines 204–210 describing the statistical model (GLM) should indicate whether covariates such as environmental factors were included in the analysis.
The results section is well organized, and the tables effectively present genotype frequencies and their association with egg production traits.
Lines 252–256: the SNP visualization figure (Figure 2) should also be annotated to highlight the functional significance of the SNP locations in the NPY gene.
The discussion effectively interprets the results but lacks strong integration with prior literature.
Lines 386–388 mention that PH had higher egg production than other native breeds, but the authors should have more discussion about whether this is due to genetic advantages, environmental conditions, or management factors.
Lines 453–468: the discussion of SNP associations with egg production traits should include comparisons to similar studies in other chicken populations.
The conclusion is well-written and concise, summarizing the study’s key findings.
Lines 490–492 should include a statement or citation on how these findings can influence breeding programs and conservation strategies for native chickens in the discussion part.
For final recommendation, the study is scientifically sound and provides valuable insights into the use of NPY gene SNP markers for egg production improvement in native chickens. However, the manuscript requires minor revisions before it is suitable for publication. If these revisions are made, the paper would be a strong candidate for publication in Animals.
Author Response
Dear Reviewer2,
Response to Reviewer 2 Comments
This study investigates the role of single nucleotide polymorphisms (SNPs) in the neuropeptide Y (NPY) gene and their association with egg production traits in Thai native, black-bone, and commercial layer chickens. The study aims to identify genetic markers that could be applied in marker-assisted selection (MAS) to improve the productivity of native chicken breeds. The study is well-structured, presenting a clear research question and appropriate methodology. However, some areas require further clarification and revision.
We sincerely appreciate the reviewer’s positive feedback, and their recognition of the improvements made in the manuscript. We hope that the revised manuscript will meet your expectations. Our responses to each comment are listed below.
Point 1: The abstract effectively summarizes the study but could better highlight the practical applications of these findings in real-world poultry breeding programs. For example, the statement "These findings highlight the potential of using genetic markers for improving egg production traits in Thai native chickens" (lines 46–48) should be expanded to explain how such genetic markers could be implemented in MAS programs in the discussion part.
Response 1: We have added a sentence in the discussion section explaining how these genetic markers could be implemented in MAS programs, as follows:
“This study provides valuable insights that contribute to the improvement of poultry breeding programs by incorporating genetic markers to enhance productivity while preserving the unique traits of native chicken breeds. The genetic markers identified in this study could be integrated into Marker-Assisted Selection (MAS) programs, allowing for the early selection of hens with superior egg production traits. This approach would improve breeding efficiency, shorten generation intervals, and enhance the productivity of Thai native chickens while maintaining their genetic diversity.” Please see lines 524-531.
Point 2: The introduction provides relevant background information on the economic and cultural significance of native chickens.
However, lines 55–56 mention the role of native chickens in circular economies without citing recent studies supporting this claim.
Response 2: We have added a sentence providing relevant background information on the economic and cultural significance of native chickens, along with related references, as follows.
“Native chickens contributed to sustainable farming by utilizing agricultural by-products, reducing food waste, and supporting smallholder livelihoods. Several recent studies highlighted their role in sustainable food systems and their contribution to circular economies within communities worldwide [1-3].” Please see lines 58-62.
References:
- Padhi, M.K. Importance of indigenous breeds of chicken for rural economy and their improvements for higher production
performance. Scientifica 2016, 2016, 2604685. https://doi.org/10.1155/2016/2604685
- Abbasi, I.A.; Shamim, A.; Shad, M.K.; Ashari, H.; Yusuf, I. Circular economy-based integrated farming system for indigenous chicken: Fostering food security and sustainability. Clean. Prod. 2024, 436, 140368. https://doi.org/10.1016/j.jclepro.2023.140368
- Loengbudnark, W.; Chankitisakul, V.; Duangjinda, M.; Boonkum, W. Sustainable growth through thai native chicken farming: lessons from rural communities. Sustainability 2024, 16, 7811. https://doi.org/10.3390/ su16177811
Point 3: Lines 66–68 should include actual production statistics for native chickens to provide a more quantitative comparison with commercial layers.
Response 3: We have revised lines 66–68 to incorporate these statistics and provide the relevant citations as follows.
“According to recent studies, native chickens typically produce 80–180 eggs per year, whereas commercial layers can produce 250–320 eggs per year under optimal conditions [3,12,13]”. Please see lines 73-75.
References:
- Chomchuen, K.; Tuntiyasawasdikul, V.; Chankitisakul, V.; Boonkum, W. Genetic Evaluation of Body Weights and Egg Production Traits Using a Multi-Trait Animal Model and Selection Index in Thai Native Synthetic Chickens (Kaimook e-san2). Animals 2022, 12, 335. https://doi.org/10.3390/ ani12030335
- Mengesha, Y., Kebede, E., & Getachew, A. (2022). Review of chicken productive and reproductive performance and its challenges in Ethiopia. All Life, 15(1), 118–125. https://doi.org/10.1080/26895293.2021.2024894
- Loengbudnark, W.; Chankitisakul, V.; Duangjinda, M.; Boonkum, W. Sustainable growth through thai native chicken farming: lessons from rural communities. Sustainability 2024, 16, 7811. https://doi.org/10.3390/ su16177811
The materials and methods section is generally well-detailed but lacks specific information on some key aspects.
Point 4: Lines 144–145 mention that blood samples were randomly collected, but it is unclear whether they were obtained from different farms or a single research station.
Response 4: We have revised the sentence of blood samples to clearly state that blood samples were obtained from multiple research stations as follows:
“Pradu Hang Dum (PH) samples were obtained from the Chiangmai Livestock Research and Breeding Center. Chee (C) and Black-Bone (BB) samples were collected from the Network Center for Animal Breeding and Omics Research, Faculty of Agriculture, Khon Kaen University. Commercial layer chicken (LC) samples were collected from the Faculty of Agricultural Technology, Kalasin University.” Please see lines 162-166.
Point 5: Lines 157–163: the DNA extraction method follows a standard protocol, but the authors should justify why guanidine hydrochloride was used instead of other commonly used DNA extraction methods.
Response 5: We used the guanidine hydrochloride method for DNA extraction because it is highly effective in isolating high-quality DNA, particularly from avian blood samples, which often contain inhibitors. This method ensures efficient cell lysis, protein denaturation, and nucleic acid stabilization, resulting in high DNA yield and purity. Compared to other commonly used methods, guanidine hydrochloride extraction is cost-effective, time-efficient, and well-suited for large-scale sample processing. We have revised the manuscript to include a justification for this choice in the methodology section. Please see lines 180-186.
“In this study, guanidine hydrochloride was chosen for DNA extraction because it is highly effective in isolating high-quality DNA, particularly from avian blood samples, which often contain inhibitors. This method ensures efficient cell lysis, protein denaturation, and nucleic acid stabilization, resulting in high DNA yield and purity. Compared to other commonly used methods, guanidine hydrochloride extraction is cost-effective, time-efficient, and well-suited for large-scale sample processing.”
Point 6: Lines 204–210 describing the statistical model (GLM) should indicate whether covariates such as environmental factors were included in the analysis.
Response 6: Thank you for your comment. In our statistical model (GLM), we accounted for key fixed effects such as breed and age of hens included in the model. Also, environmental factors and management were included as covariates in the model. We have revised the sentence to clarify whether environmental factors were included in the analysis as follows:
“The egg production traits were analyzed using the Proc GLM procedure. The model included fixed effects such as breed and age of hens, along with covariates for environmental factors and management.” Please see lines 227-229.
The results section is well organized, and the tables effectively present genotype frequencies and their association with egg production traits.
Point 7: Lines 252–256: the SNP visualization figure (Figure 2) should also be annotated to highlight the functional significance of the SNP locations in the NPY gene.
Response 7: We have added the sentence about an explanation highlighting the significance of SNPs in the NPY gene as follows:
“The identification of SNPs within the coding region of the NPY gene is particularly significant, as NPY plays a crucial role in regulating appetite, energy balance, and reproductive functions in poultry. Variations in these loci may influence egg production traits by modulating hormonal pathways associated with ovarian activity and metabolism. Therefore, these SNPs serve as valuable genetic markers for future research on selective breeding strategies aimed at enhancing the productivity of Thai native chickens and related breeds.” Please see lines 284-290.
The discussion effectively interprets the results but lacks strong integration with prior literature.
Point 8: Lines 386–388 mention that PH had higher egg production than other native breeds, but the authors should have more discussion about whether this is due to genetic advantages, environmental conditions, or management factors.
Response 8: We have added the sentence about PH had higher egg production than other native breeds and reasons due to genetic advantages, environmental conditions, or management factors as follows:
“Among Thai native breeds, the Pradu Hang Dum (PH) chickens exhibited higher productivity than the Chee (C) chickens, particularly in number of eggs at 9 and 12 months (NE_9M ,NE_12M) of egg production period, number of eggs per month (EperM) and egg mass (EMs) traits. This difference can be attributed to the fact that PH chickens have been genetically selected for egg production traits over a period of more than 15 years, leading to improved genetic potential for these traits. In contrast, C chickens have primarily been raised for genetic conservation, without any explicit selection for egg production traits. Consequently, egg production traits in C chickens have not been as effectively selected or improved. Additionally, differences in environmental conditions and management practices also influenced egg production.” Please see lines 421-430.
Point 9: Lines 453–468: the discussion of SNP associations with egg production traits should include comparisons to similar studies in other chicken populations.
Response 9: We have included comparisons to similar studies in other chicken populations as follows:
“According to previous studies, Chen et al. [40] identified 12 significant SNPs associated with Age at First Egg (AFE) in layer breeds, which showed strong correlations with body weight and first egg weight. However, there were no overlapping genes between breeds, indicating distinct genetic foundations for AFE. In addition, Fu et al. [41] reported 11 genome-wide significant SNPs related to egg production traits, including age at first egg and egg number. Notably, SNPs in the NEO1, ADPGK, and CYP11A1 genes were linked to egg number, which could aid in selective breeding strategies for chickens. Zhou et al. [42] identified 9 significant SNP loci associated with egg production traits in Peking ducks, with SNPs linked to age at first egg and egg number. These SNPs showed varying correlations with traits, suggesting their potential for use in selective breeding strategies. Furthermore, other studies have highlighted the influence of the NPY gene on egg production in Indigenous Noi chickens [26].” Please see lines 509-521.
References:
- Chen, A.; Zhao, X.; Wen, ; Zhao, X.; Wang, G.; Zhang, X.; Ren, X.; Zhang, Y.; Cheng, X.; Yu, X.; Mei, X.; Wang, H.; Guo, M.; Jiang, X.; Wei, G.; Wang, X.; Jiang, R.; Guo, X.; Ning, Z.; Qu, L. Genetic parameter estimation and molecular foundation of chicken egg-laying trait. Poult. Sci. 2024, 103, 103627. https://doi.org/10.1016/j.psj.2024.103627
- Fu, M.; Wu, Y.; Shen, J.; Pan, A.; Zhang, H.; Sun, J.; Liang, Z.; Huang, T.; Du, J.; Pi, J. Genome-wide association study of egg production traits in shuanglian chickens using whole genome sequencing. Genes 2023, 14, 2129. https://doi.org/ 10.3390/genes14122129
- Zhou, J.; Yu, J.-Z.; Zhu, M.-Y.; Yang, F.-X.; Hao, J.-P.; He, Y.; Zhu, X.-L.; Hou, Z.-C.; Zhu, F. Genome-wide association analysis and genetic parameters for egg production traits in Peking ducks. Animals 2024, 14, 1891. https:// doi.org/10.3390/ani14131891
- Ngu, N.T.; Xuan, N.H.; Vu, C.T.; An, N.T.; Dung, T.N.; Nhan, N.T.H. Effects of genetic polymorphisms on egg production in indigenous Noi chicken. Exp. Biol. Agric. Sci. 2015, 3, 487–493.
The conclusion is well-written and concise, summarizing the study’s key findings.
Point 10: Lines 490–492 should include a statement or citation on how these findings can influence breeding programs and conservation strategies for native chickens in the discussion part.
Response 10: We have added the sentence related the importance of breeding programs and conservation strategies for native chickens in the discussion part as follows:
“This study provides valuable insights that contribute to the improvement of poultry breeding programs by incorporating genetic markers to enhance productivity while preserving the unique traits of native chicken breeds. The genetic markers identified in this study could be integrated into Marker-Assisted Selection (MAS) programs, allowing for the early selection of hens with superior egg production traits. This approach would improve breeding efficiency, shorten generation intervals, and enhance the productivity of Thai native chickens while maintaining their genetic diversity.
In addition, by utilizing the genetic diversity within native populations, breeding programs can select for traits that improve egg production, disease resistance, and adaptability to local conditions, leading to more efficient and sustainable poultry farming. Furthermore, such programs help conserve native breeds by ensuring their continued relevance in agricultural systems, providing both economic benefits and the protection of valuable genetic resources. This approach aligns with the dual goal of improving productivity while safeguarding the genetic heritage of native poultry breeds.” Please see lines 526-540.
Point 11: For final recommendation, the study is scientifically sound and provides valuable insights into the use of NPY gene SNP markers for egg production improvement in native chickens. However, the manuscript requires minor revisions before it is suitable for publication. If these revisions are made, the paper would be a strong candidate for publication in Animals.
Response 11: We are grateful for your critical reading and efforts to improve the quality of the manuscript. We hope that the revised manuscript will meet your expectations. We have addressed all the reviewers' comments and revised the manuscript accordingly to ensure the strong research for publication in Animals.
Best Regards,
Wuttigrai Boonkum

Reviewer 3 Report
Comments and Suggestions for Authors
This study aimed to identify single nucleotide polymorphism (SNP) loci within the coding sequence of the neuropeptide Y (NPY) gene and evaluate their association with egg production traits in Thai native chickens. The results obtained are important for improving egg production rates as a result of breeding selection. The research methods used are correct. Sufficient discussion. References selection correct and well used.
General comments:
In my opinion, the article should be supplemented with the following information:
In the Materials and methods chapter there is no information about:
- Type of building – closed buliding, windowless, method of kept birds - the cage (single or group) or litter system?
- Temperature, relative humidity, light length and intensity.
- No information about the type of scale for recording of egg weight and measurement accuracy
Others
Add spaces when specifying the temperature, for example 4 °C not 4°C
For significance, use a low letter "p" in italic (p < 0.05) instead of the "P" in the main article and space without spaces
Please use a "dot" after each abbreviation, for example "Sys. Rev. Pharm." instead of Sys Rev Pharm.
Detailed comments
L4 laying hens instead of layers. Layers are also female ducks, geese, turkeys during egg production
L37 9 months of egg production period instead of current form
L38 months of egg production period
LL44 (PIC = 0.22-0.50) according to L330
L86 [11,12]
L142 Please provide the trade name commercial laying hens breed
L146-147 What does the term first egg weight mean - is it the mass of eggs collected on the first day of egg production or in the first week of egg production? How was the egg mass determined, how was the egg weight (type of weight) controlled, was it individually or collectively?
L147 weight with a lower case letter
L149 These traits, or what? First egg weight too? Or just the number of egg and egg weight?
L160, 161 4 °C, space after 4
L196+ What was the experimental unit, what level of "p" was used to verify differences?
L220 196.90 or 196.91?
L221, 226 (p < 0.001)
L233 egg weight instead of weight?
In Table 1, EMs for LC = 60.67 g, did the hens lay more than 1 egg per day?, because according to Table 1 EW 12M was 53.5 g, while the number of eggs in 12 months (365 days) was 254.44 eggs, which was less than 1 egg per day.
L303 0.58 or 0.56?
L356 AFEP for GGab or GGb as described?
In Table 3 for SNP4, SNP5, SNP6 where p-value is 0.00 please change to "<0.01"
L419 Promket et al. [33] instead of current form
L420 Ding et al. [34]
L480 Pradu Hang Dum breed, please add “breed”
L537 One instead of ONE
L548 Sys. Rev. Pharm.
L564 BMC Genom. instead of BMC Genomics
L569 Earth Environ. Sci.
L586 J. Anim. Sci.
L591 J. Anim. Sci.
Author Response
Dear Reviewer3,
Response to Reviewer 3 Comments
Point 1: This study aimed to identify single nucleotide polymorphism (SNP) loci within the coding sequence of the neuropeptide Y (NPY) gene and evaluate their association with egg production traits in Thai native chickens. The results obtained are important for improving egg production rates as a result of breeding selection. The research methods used are correct. Sufficient discussion. References selection correct and well used.
Response 1: We sincerely appreciate the reviewer’s positive feedback and their recognition of the improvements made in the manuscript. We hope that the revised manuscript will meet your expectations. Our responses to each comment are listed below.
General comments:
In my opinion, the article should be supplemented with the following information:
In the Materials and methods chapter there is no information about:
Point 2: Type of building – closed building, windowless, method of kept birds - the cage (single or group) or litter system?
Response 2 We added the following sentences to clarify the housing conditions: “All hens were reared in an open-housing system with a window design, receiving an average of 12 hours of natural light per day.” and “…At 16 weeks of age, the hens were transferred to single battery cages (20 × 45 × 40 cm)…” Please see lines 153-155 and 156-157.
Point 3: Temperature, relative humidity, light length and intensity.
Response 3: We added the following sentences to clarify the temperature, relative humidity, light length and intensity conditions: “…, receiving an average of 12 hours of natural light per day. The daily air temperature was maintained between 22–36°C, and relative humidity ranged from 50–80% throughout the study period.” Please see lines 154-156.
Point 4: No information about the type of scale for recording of egg weight and measurement accuracy
Response 4: We added the following sentence to clarify the type of scale used for recording egg weight and its measurement accuracy: “Egg weight was measured using a digital electronic scale with a precision of 0.01 g.” Please see lines 172-173.
Others
Point 5: Add spaces when specifying the temperature, for example 4 °C not 4°C
Response 5: We revised the text as you suggested. Please see lines 187 and 189.
Point 6: For significance, use a low letter "p" in italic (p < 0.05) instead of the "P" in the main article and space without spaces
Response 6: We revised the letter “P” was corrected to '(p < 0.001)' throughout the revised manuscript. Please see lines 254 and 259.
Point 7: Please use a "dot" after each abbreviation, for example "Sys. Rev. Pharm." instead of Sys Rev Pharm.
Response 7: We revised the letter " Sys. Rev. Pharm." instead of "Sys Rev Pharm." in the manuscript as per the suggestion. Please see line 620.
Detailed comments
Point 8: L4 laying hens instead of layers. Layers are also female ducks, geese, turkeys during egg production
Response 8: We corrected the word to “layer hens” as you suggested. Please see line 4.
Point 9: L37 9 months of egg production period instead of current form
Response 9: We revised the word to “9 months (EW_9M) and 12 months (EW_12M) of the egg production period” Please see lines 37–38.
Point 10: L38 months of egg production period
Response 10: We revised the word to “9 months (NE_9M) and 12 months (NE_12M) of egg production period” Please see lines 38–39.
Point 11: L44 (PIC = 0.22-0.50) according to L330
Response 11: We revised the word to “0.22 to 0.50”. Please see line 45.
Point 12: L86 [11,12]
Response 12: The reference has revised to “[14,15]”. Please see line 93.
Point 13: L142 Please provide the trade name commercial laying hens breed
Response 13: We provided the trade name of the commercial laying hen breed to “commercial laying hens of the Hy-Line Brown breed”. Please see line 153.
Point 14: L146-147 What does the term first egg weight mean - is it the mass of eggs collected on the first day of egg production or in the first week of egg production? How was the egg mass determined, how was the egg weight (type of weight) controlled, was it individually or collectively?
Response 14: First egg weight (FEW) refers to the mass of the first egg laid by a hen collected on the first day of egg production for each bird. In this study, we measured the weight of each hen’s first egg on the day it was laid. Each egg was weighed individually using a calibrated digital scale to ensure accuracy. To minimize variability, all eggs were weighed at the same time each day following a standardized procedure.
Point 15: L147 weight with a lower-case letter
Response 15: We revised the word “weight” with a lower-case letter. Please see line 168.
Point 16: L149 These traits, or what? First egg weight too? Or just the number of egg and egg weight?
Response 16: To improve clarity, we have revised the term “egg per month (EperM)” to “number of eggs per month (EperM). Please see lines 170-171.
For egg mass (EMs), we assessed hen productivity and laying performance using the following calculation:
. Meanwhile, egg weight was used to analyze egg size, quality, and breed differences. It was measured using a digital scale in grams per egg. Therefore, this study included both egg weight and egg mass as important parameters.
Point 17: L160, 161 4 °C, space after 4
Response 17: We already revised. Please see lines 187 and 189.
Point 18: L196+ What was the experimental unit, what level of "p" was used to verify differences?
Response 18: We provided the experimental unit and level of "p" was used to verify differences. Please see in lines 224-225, and 230.
Point 19: L220 196.90 or 196.91?
Response 19: The NE_12M number was corrected to 196.90±43.32 eggs. Please see line 252.
Point 20: L221, 226 (p < 0.001)
Response 20: The p-value was corrected to (p < 0.001). Please see lines 254 and 259.
Point 21: L233 egg weight instead of weight?
Response 21: The word “weight” was corrected to “egg weight”. Please see line 266.
Point 22: In Table 1, EMs for LC = 60.67 g, did the hens lay more than 1 egg per day?, because according to Table 1 EW 12M was 53.5 g, while the number of eggs in 12 months (365 days) was 254.44 eggs, which was less than 1 egg per day.
Response 22: The egg mass (Ems) value of 60.67 g for LC was calculated using the formula:
Since the average egg weight at 12 months (EW_12M) was 53.5 g, but the EMs was 60.67 g, this suggests that egg weight varied over time. Additionally, the total number of eggs laid in 12 months (254.44 eggs) was less than one egg per day on average (0.70 eggs per day). Therefore, the hens did not lay more than one egg per day. The slightly higher EMs value may be due to variation in egg weight at different time points in the laying cycle.
Point 23: L303 0.58 or 0.56?
Response 23: In Table 2, the GG genotype had the lowest value, recorded at 0.56.
Point 24: L356 AFEP for GGab or GGb as described?
Response 24: In AFEP trait, GG ab was corrected. However, to improve the clarity of the results, we have revised the sentence as follows:
“SNP4 (g.404 G>T) was significantly associated with AFEP and first egg weight (FEW). Hens with the TT genotype had the lowest AFEP (156.23 days), whereas those with the GT genotype had the highest AFEP (168.63 days) (p < 0.05). For the FEW trait, hens with the GT genotype had the highest FEW compared to other genotypes (p < 0.05).” Please see lines 392-395.
Point 25: In Table 3 for SNP4, SNP5, SNP6 where p-value is 0.00 please change to "<0.01"
Response 25: We have revised the p-values for SNP4, SNP5, and SNP6 from “0.00” to “<0.01” as you suggested. Please see the revised Table 3.
Point 26: L419 Promket et al. [33] instead of current form
Response 26: We have revised the reference as you suggested. Please see line 460.
Point 27: L420 Ding et al. [34]
Response 27: We have revised the reference as you suggested. Please see line 461.
Point 28: L480 Pradu Hang Dum breed, please add “breed”
Response 28: We have revised the text as you suggested. Please see line 547.
Point 29: L537 One instead of ONE
Response 29: We have revised the text as you suggested. Please see line 604.
Point 30: L548 Sys. Rev. Pharm.
Response 30: We have revised the text as you suggested. Please see line 620.
Point 31: L564 BMC Genom. instead of BMC Genomics
Response 31: We have revised the text as you suggested. Please see line 636.
Point 32: L569 Earth Environ. Sci.
Response 32: We have revised the text as you suggested. Please see line 643.
Point 33: L586 J. Anim. Sci.
Response 33: We have revised the text as you suggested. Please see line 658.
Point 34: L591 J. Anim. Sci.
Response 34: We have revised the text as you suggested. Please see line 663.
Best Regards,
Wuttigrai Boonkum

Round 2
Reviewer 3 Report
Comments and Suggestions for Authors
This study aimed to identify single nucleotide polymorphism (SNP) loci within the coding sequence of the neuropeptide Y (NPY) gene and evaluate their association with egg production traits in Thai native chickens. The results obtained are important for improving egg production rates as a result of breeding selection. The research methods used are correct. Sufficient discussion. References selection correct and well used.
General comments:
The revised version of this article includes responses to comments in the previous review.
Detailed comments:
In Table 1, EMs for LC = 60.67 g, did the hens lay more than 1 egg per day?, because according to Table 1 EW 12M was 53.5 g, while the number of eggs in 12 months (365 days) was 254.44 eggs, which was less than 1 egg per day.
L170 number of number?
Author Response
Dear Reviewer 2,
Response to Reviewer 2 Comments
This study aimed to identify single nucleotide polymorphism (SNP) loci within the coding sequence of the neuropeptide Y (NPY) gene and evaluate their association with egg production traits in Thai native chickens. The results obtained are important for improving egg production rates as a result of breeding selection. The research methods used are correct. Sufficient discussion. References selection correct and well used.
We sincerely appreciate the reviewer’s positive feedback, and their recognition of the improvements made in the manuscript. We hope that the revised manuscript will meet your expectations. Our responses to each comment are listed below.
General comments:
The revised version of this article includes responses to comments in the previous review.
Detailed comments:
Point 1: In Table 1, EMs for LC = 60.67 g, did the hens lay more than 1 egg per day?, because according to Table 1 EW 12M was 53.5 g, while the number of eggs in 12 months (365 days) was 254.44 eggs, which was less than 1 egg per day.
Response 1: From our data including EMs for LC = 60.67 g, EW_12M (Average Egg Weight over 12 Months) = 53.5 g, NE_12M (Number of eggs in 12 months) = 254.44 eggs, and A year has 365 days. To determine if the hens laid more than 1 egg per day, check whether the average daily egg production exceeds 1. Average daily egg production = NE_12M / 365 = eggs per day. Since 0.697 eggs per day is less than 1 egg per day, the hens did not lay more than one egg per day. However, if use another calculation method by Ems/EW = eggs per day. This value may suggest that egg production appears greater than 1 per day when using the egg mass formula, but we do not use this calculation formula in our analysis.
Point 2: L170 number of number?
Response 2: We have deleted the word “number of” from the revised manuscript. Please see line 170.
Best Regards,
Wuttigrai Boonkum
